# Impact of *BRAF*V600E Mutation on Event-Free Survival in Patients with Papillary Thyroid Carcinoma: A Retrospective Study in a Romanian Population

**DOI:** 10.3390/cancers15164053

**Published:** 2023-08-11

**Authors:** Adela Nechifor-Boilă, Ancuţa Zahan, Claudia Bănescu, Valeriu Moldovan, Doina Piciu, Septimiu Voidăzan, Angela Borda

**Affiliations:** 1Department of Histology, Center for Advanced Medical and Pharmaceutical Research, George Emil Palade University of Medicine, Pharmacy, Science and Technology of Targu-Mureș, 38th Gh. Marinescu Street, 540139 Targu Mures, Romania; ancutza_ct@yahoo.com (A.Z.); angela.borda@umfst.ro (A.B.); 2Department of Pathology, Targu-Mures Clinical County Hospital, 28 First December 1918 Street, 540061 Targu Mures, Romania; 3Department of Genetics, Center for Advanced Medical and Pharmaceutical Research, George Emil Palade University of Medicine, Pharmacy, Science and Technology of Targu-Mureș, 38th Gh. Marinescu Street, 540139 Targu Mures, Romania; claudia.banescu@umfst.com (C.B.); valeriumoldovan@gmail.com (V.M.); 4Department of Nuclear Medicine, “Ion Chiricuţă” Institute of Oncology, 400015 Cluj-Napoca, Romania; doina.piciu@gmail.com; 5Department of Epidemiology, George Emil Palade University of Medicine, Pharmacy, Science and Technology of Targu-Mureș, 38th Gh. Marinescu Street, 540139 Targu Mures, Romania; septimiu.voidazan@umfst.ro; 6Department of Pathology, Targu-Mureș Emergency County Hospital, 50 Gh. Marinescu Street, 540139 Targu Mures, Romania

**Keywords:** papillary thyroid cancer, *BRAF*V600E mutation, lymph node metastasis, prognosis, outcomes

## Abstract

**Simple Summary:**

Although papillary thyroid carcinoma (PTC) is generally a highly curable disease, there is a small subgroup of PTC cases that behaves more aggressively, with high rates of disease recurrence, tumor progression and distant metastases. These patients need to be accurately identified for an appropriate, more-aggressive therapeutical approach. *BRAF*V600E mutation is the most prevalent genetic event in PTC. However, its role as a prognostic factor remains unclear. Herein, we aimed to assess the prognostic value of *BRAF*V600E mutation as a single factor, as well as in synergic interaction with other demographic and pathological risk factors in a series of 127 PTCs. Our results highlight a significant impact of *BRAF*V600E mutation on event-free survival among PTC patients. Nevertheless, *BRAF*V600E mutation status should not be used as an independent predictive factor in PTC patients, but rather should be integrated in the context of other clinicopathological risk factors.

**Abstract:**

We aimed to evaluate the prognostic value of *BRAF*V600E mutation in a series of 127 papillary thyroid carcinoma (PTC) cases as a single factor, and in synergic interaction with other standard risk factors. *BRAF*V600E mutation was assessed by real-time PCR. Event-free survival (EFS) was calculated between the date of the first evaluation and the date of occurrence of an adverse event or the date of the last known status. The prevalence of *BRAF*V600E mutation was 57.2%. The Kaplan–Meier analysis showed a significant reduction of EFS among cases harboring *BRAF*V600E mutation compared to non-mutated cases (*p* = 0.010). In addition, *BRAF*V600E mutation was found to better predict adverse outcomes when associated with the following risk factors: age ≥ 55 years old (*p* < 0.001), male gender (*p* < 0.001), conventional (*p* = 0.005) and tall cell (*p* = 0.014) histology, tumor size > 40 mm (*p* = 0.001), extrathyroidal extension (*p* = 0.001), multifocality (*p* = 0.001) and lymph node metastasis (*p* < 0.001). In univariate analysis, a 3.74-fold increased risk for a reduced EFS (*p* = 0.018) was found for *BRAF*V600E-mutated cases, but no increased risk was further confirmed by multivariate analysis. Our results highlight that *BRAF*V600E mutation cannot be used alone as an independent predictive factor in PTC patients, but is prognostically valuable if integrated in the context of other clinicopathological risk factors.

## 1. Introduction

Papillary thyroid carcinoma (PTC) is the most common endocrine malignancy, accounting for 80% of all thyroid cancers [1]. Worldwide, the incidence of PTC has significantly increased over the last 30 years [2,3], yet with no increase in the mortality rate [4].

Although PTC is generally a highly curable disease, there is a small subgroup of PTC cases that tends to behave aggressively, with high rates of disease recurrence, tumor progression or even distant metastases leading to poor prognosis [5,6]. These patients need to be accurately identified for an appropriate, more-aggressive therapeutical approach to reduce the chance of disease recurrence and worse outcomes. Moreover, patient’s quality of life is of paramount importance, so it is not only important to aggressively treat an aggressive cancer, but also to alleviate the physiological burden of an indolent one [5].

In recent years, the development of targeted therapies has led to increased interest in the identification of molecular alterations present in thyroid cancer and their prognostic impact [7]. *BRAF*V600E mutation has received the widest attention, being by far the most prevalent genetic event in patients with PTC, with a reported prevalence of 25–82.3% [8]. *BRAF*V600E mutation is caused by a thymine to adenine transversion at nucleotide 1799 (T1799A) [9], leading to a substitution of Valine by Glutamic acid at residue 600 of the protein (V600E). A result of the genetic alteration is the activation of the mitogen-activated protein kinase (MAPK) signaling pathway [10], which plays a major role in the regulation of cell growth, division and proliferation [11]. Many studies have demonstrated an association of *BRAF*V600E mutation with aggressive clinicopathologic characteristics of PTC, showing promise of this mutation as a prognostic molecular marker for PTC [8,12,13,14,15,16]. Nevertheless, literature data is controversial and the prognostic value of *BRAF*V600E mutation has been questioned, with other studies failing to demonstrate that *BRAF*V600E is an independent prognostic factor for PTC [17,18]. Moreover, the prevalence of *BRAF*V600E mutation in PTCs is high, reaching up to 80% in some studies [8], whereas the frequency of negative outcome for PTC patients is low (10–15%) [17,18]. Hence, if we consider *BRAF*V600E mutation in isolation as a single prognostic factor, the risk of over- or undertreatment would be considerable for many PTC patients [6]. Therefore, *BRAF*V600E mutation should be evaluated together with other prognostic factors [6]. In the study performed by Gan X. et al. [19], *BRAF*V600E mutation was found to better predict aggressive and recurrent PTC based on age stratification with a cut-off age of 55 years old. The authors concluded that synergic interaction between *BRAF*V600E mutation and age stratification may help clinicians in terms of optimal decision-making regarding surgical approach and extent of surgery.

In the present study, we first evaluated the relationship between *BRAF*V600E mutational status and demographic, pathological and outcome characteristics of PTC patients in a series of 127 cases. Further on, we aimed to assess the prognostic value of *BRAF*V600E mutation in our series of PTC cases, as a single factor, as well as in synergic interaction with other standard demographic and pathological risk factors.

## 2. Materials and Methods

### 2.1. Case Selection

All consecutive PTC cases registered at the Pathology Department, Targu-Mureş Emergency Hospital, Romania, between 2008–2015 were evaluated. Criteria for inclusion in the study were: (1) a histopathological diagnosis consistent with PTC; (2) tumor size of at least 10 mm; (3) availability of hematoxylin/eosin (HE)-stained slides for case review; (4) well-preserved formalin-fixed paraffin-embedded (FFPE) tumor blocks of the corresponding cases available in the archive for molecular assay; and (5) available follow-up data.

### 2.2. Pathological Data

The corresponding HE-stained slides for all the cases included in the study were reviewed by two endocrine pathologists (ANB and AB). Tumor histology and pathological stage were reassessed according to the 2017 WHO (World Health Organization) Classifications of Tumors of the Thyroid Gland [20] (pp. 81–91) and the 2017 American Joint Committee on Cancer/Union for International Cancer Control (AJCC/UICC) TNM Classification of Tumors [21] (pp. 87–96). All cases with controversial features were discussed and a consensus was reached using a multi-headed microscope.

The diagnosis of PTC relied on tumor’s architecture (either papillary or follicular pattern with invasive characteristics) and nuclear features (enlargement, overlapping, nuclear contours’ irregularity, grooves, clearing, nuclear pseudoinclusions).

The following demographic and pathological features with prognostic significance were evaluated: (1) *patients’ age at diagnosis* with cut-off values of 55 years old; (2) *patients’ gender*; (3) *tumor size* and (4) *histological type* (conventional, or variant of PTC such as follicular, tall-cell, Warthin-like, oncocytic or solid); (5) *extrathyroidal extension,* defined as tumor extension into strap muscles (sternohyoid, sternothyroid or omohyoid muscles); (6) *multifocality,* defined as the presence of two or more isolated/non-contiguous tumor foci in one or both thyroid lobes; (7) *lymph node metastasis*, defined as involvement of at least one regional lymph node; (8) *surgical resection margins status*; (9) *vascular invasion* and (10) *stage grouping*.

### 2.3. Molecular Analysis

For each case, one representative FFPE block was selected for the molecular assay. The selected FFPE block corresponded to well-preserved, high-density tumor areas, with an absence of hemorrhage and calcifications. The area of interest (the tumor area) was circled on the HE-stained slides. Using the HE-stained slide as a guide and a standard microscope, a manual microdissection of the marked area was performed. DNA isolation was accomplished using MasterPure^TM^ DNA purification kit (Epicentre, Madison), as previously described [22]. All real-time PCR experiments were performed at the Platform of Molecular Biology, Center for Advanced Medical and Pharmaceutical Research, George Emil Palade University of Medicine, Pharmacy, Science and Technology of Târgu-Mureș, using a 7500 Fast Dx RT-PCR Instrument (Applied Biosystems, Waltham, Massachusetts, USA). The Thyroid Cancer Mutation Analysis Kit (EntroGen, Woodland Hills, California, USA) was used for the detection of the somatic *BRAF*V600E mutation.

### 2.4. Follow-Up Data

Follow-up covered the period between January 2001 and December 2017; it was defined as the period between the initial surgical treatment and the last clinical evaluation. We collected data from the Department of Nuclear Medicine, “Ion Chiricuţă” Institute of Oncology, Cluj-Napoca, Romania, where all patients included in our study were later referred to for adjuvant treatment (^131^I ablation) and follow-up, following the surgical treatment.

The 2015 American Thyroid Association (ATA) risk of recurrence stratification system [23] was used to determine the disease status, based on follow-up data available at the last clinical evaluation. A *disease-free status* was characterized by the absence of detectable residual disease (on ultrasound and whole-body scans (WBS)) and low basal (<0.2 ng/mL) and stimulated (<1 ng/mL) thyroglobulin (Tg) serum levels. *Persistent disease* was considered when there was evidence of a detectable residual or metastatic tumor (on ultrasound, WBS, CT (Computed Tomography) and ^18^FDG-PET-CT (Positron Emission Tomography with 2-deoxy-2-[fluorine-18] fluoro-d-glucose integrated with Computed Tomography)) and/or in case of elevated basal (>0.2 ng/mL) and stimulated (>1 ng/mL) Tg serum levels. The appearance of a new biochemical disease or tumor recurrence in patients previously classified as disease free were typical defining characteristics for *recurrent disease*. Secondary, metastatic tumors identified at the time of diagnosis or during the follow-up period qualified as *distant metastases*.

### 2.5. Statistical Analysis

Statistical analysis was performed using the Statistical Package for Social Sciences (SPSS, version 20, Chicago, IL, USA). Data were labeled as nominal or quantitative variables. Nominal variables were expressed as number and percentages and were compared using the chi-squared test or Fisher’s exact test (when the conditions of application of chi-square test were not met).

Quantitative variables were tested for normality of distribution using the Kolmogorov–Smirnov test, graphically confirmed with a histogram, and were described by mean ± standard deviation or median and percentiles (25; 75%), whenever appropriate. The Student’s *t* test was applied to compare continuous values with Gaussian distribution.

Survival curves were obtained using a Kaplan–Meier model and compared using the long-rank test. Persistent disease, recurrent disease, or distant metastases occurring during the follow-up period were considered as adverse events. Event-free survival (EFS) was calculated between the date of the first evaluation and the date of occurrence of an adverse event or the date of the last known status.

Prognostic factors of adverse events were determined using a Cox model after assessment of the proportionality of risk hypothesis, first in univariate analysis, and if appropriate in multivariate analysis, including factors found significant in univariate analysis.

All *p*-values were two-sided, and a *p* < 0.05 was considered to indicate statistically significant differences.

## 3. Results

### 3.1. Patient’s Characteristics

Our study included 127 patients (110 females and 17 males; mean ± standard deviation [SD] age 48.6 ± 1.28 years). Demographic, pathological and follow-up data for the study cases are illustrated in Table 1.

More than half of the patients included in the study were younger than 55 years old (*n* = 79, 62.2%); the mean tumor size was 22.88 ± 1.5 mm. Most of the cases were conventional PTCs (*n* = 88, 69.3%), while PTC variants were less numerous (tall cell *n* = 9, 7.1%; Warthin-like *n* = 7, 5.5%; oncocytic *n* = 3, 2.4%; solid *n* = 2, 1.6%; follicular, infiltrative *n* = 13, 10.2%; follicular, encapsulated, invasive *n* = 5, 3.9%). Extrathyroidal extension was documented in 18.9% (*n* = 24) cases; 40.2% (*n* = 51) PTCs were multifocal. Lymph node dissection was performed in 39 (30.7%) PTCs. Of these, 26 cases displayed lymph node involvement. With regard to the primary tumor (T), 51 (40.2%), 44 (34.6%), 8 (6.3%) and 24 (18.9%) PTCs were pT1b, pT2, pT3a and pT3b, respectively. Most of the cases included in the study were stage I PTCs (79.5%, *n* = 101).

The mean follow-up period was 57 months (CI: 9–130). All patients were treated with total thyroidectomy or total thyroidectomy with lymph node dissection, and all received radioactive iodine (I^131^) therapy. At the last clinical assessment, most of the patients had a disease-free status (84.3%, *n* = 107). A persistent disease status was observed only in 14 (11%) patients; recurrence was also rare, found in only 6 (4.7%) PTC cases. Four patients with conventional PTCs and 1 with tall cell variant of PTC developed distant metastases during the follow-up period (all in the lung, and one case also in the bone).

### 3.2. Prevalence of BRAFV600E Mutation and Relationship with Demographic, Pathological and Outcomes Characteristics

The prevalence of *BRAF*V600E mutation in our study was 57.2% (67/127). The demographic, pathological and outcomes characteristics of the study cases were compared between PTCs harboring *BRAF*V600E mutation and those without (Table 1). Our data showed that *BRAF*V600E mutation was strongly associated with age ≥ 55 years old (*p* = 0.037), male gender (*p* = 0.035), conventional histology (*p* < 0.0005), extrathyroidal extension (*p* = 0.004), pT3b tumor stage (*p* = 0.007), lymph node metastasis (*p* = 0.001) and positive surgical resection margins. With regard to patient’s outcomes, although most of the patients included in our study revealed a disease-free status at the last clinical assessment, patients without *BRAF*V600E mutation were significantly more likely to be disease free (*p* = 0.008). By contrast, a persistent disease status was significantly more prevalent among PTC patients with tumors harboring *BRAF*V600E mutation (*p* = 0.009). Furthermore, all five PTC cases that developed distant metastasis during the follow-up period were *BRAF*V600E mutated (*p* = 0.031).

In univariate analysis, the presence of *BRAF*V600E mutation was associated with age ≥55 years old (*p* = 0.039), male gender (*p* = 0.043), conventional histology (*p* = 0.008), extrathyroidal extension (*p* = 0.006), lymph node metastasis (*p* < 0.001) and positive resection margins (*p* = 0.028). In multivariate analysis, conventional histology, extrathyroidal extension and lymph node metastasis remained significantly associated with the mutation (Table 2).

### 3.3. Predictive Factors

The Kaplan–Meier analysis revealed a significant impact of *BRAF*V600E mutation on EFS among our study cases. EFS at 60 months was documented in only 62.4% [CI: 54.2–70.6] PTC patients with tumors harboring *BRAF*V600E mutation compared to 91.7% [CI: 87.7–95.7] PTC patients with wild-type *BRAF*V600E tumors (long-rank test, *p* = 0.010) (Figure 1).

Moreover, our data showed that concurrent presence of *BRAF*V600E mutation with various demographic and pathological features bearing prognostic value increases the risk for a reduced EFS even more. The results are illustrated in Table 3 and Figure 2. The percentage of patients with EFS at 60 months was significantly reduced when *BRAF*V600E mutation was associated with age ≥ 55 years old (*p* < 0.001), male gender (*p* < 0.001), conventional (*p* = 0.005) and tall cell histology (*p* = 0.014), tumor size > 40 mm (*p* = 0.001), extrathyroidal extension (*p* = 0.001), multifocality (*p* = 0.001) and lymph node metastasis (*p* < 0.001).

Nevertheless, *BRAF*V600E mutation was also significantly associated with worse outcomes among lower-risk groups of PTC patients. Patients aged <55 years old (*p* < 0.021), female patients (*p* = 0.022), patients with tumors measuring ≤40 mm (*p* = 0.034) or with a single tumor focus (*p* = 0.023) but harboring *BRAF*V600E mutation revealed a significantly lower EFS compared to same PTC patients with non-*BRAF*V600E-mutated tumors (see Table 3), data further supporting the prognostic value of *BRAF*V600E mutation.

In univariate analysis, a 3.74-fold increased risk for a reduced EFS (95%CI: [1.25–11.21], *p* = 0.018) was found for *BRAF*V600E-mutated cases. A reduced EFS was also associated with age at surgery equal to 55 years old or above (*p* = 0.027), male gender (*p* = 0.005), extrathyroidal extension (*p* = 0.016), multifocality (*p* = 0.027) and lymph node metastasis (*p* < 0.001). In multivariate analysis, male gender and lymph node metastasis remained significantly associated with worse EFS (Table 4).

## 4. Discussion

*BRAF*V600E mutation represents a very specific marker for PTC, also referred to as the “genetic signature of PTC” [24,25,26]. As this mutation appears to play an important role in PTC tumorigenesis, it has been postulated that it might also have a prognostic value. Nevertheless, whether *BRAF*V600E mutation relates to more aggressive clinicopathologic features and worse outcomes in PTC patients remains variable and controversial, as highlighted by many different studies over the time [5,8,12,13,15,18,19,27,28,29].

The 2015 ATA Management Guidelines for Adult Patients with Thyroid Nodules and Differentiated Thyroid Cancer [23] emphasized that *BRAF*V600E mutational status, although not routinely recommended for initial postoperative risk stratification in differentiated thyroid cancer, has the potential to refine risk estimates when interpreted in the context of other clinicopathologic risk factors. Therefore, it appears that *BRAF*V600E mutation in isolation is not sufficient to substantially contribute to risk stratification, but an incremental improvement can be achieved if synergic interaction between *BRAF*V600E mutation and other risk factors is considered.

In the present study we evaluated the prevalence of *BRAF*V600E mutation and its relationship with demographic, pathological and outcome characteristics in a series of PTC patients. Further on, we assessed the prognostic value of *BRAF*V600E mutation in our series of cases, first as a single factor, and then in synergic interaction with other demographic and pathological risk factors.

In our study, *BRAF*V600E mutation was positive in 57.2% of PTC cases; it was found to be strongly associated with adverse demographic and pathological features, such as older age, ≥55 years old, male gender, conventional histology, extrathyroidal extension, pT3b tumor stage, lymph node metastasis, positive surgical resection margins, a persistent disease status and distant metastases. The univariate analysis confirmed these results, while in multivariate analysis, conventional histology, extrathyroidal extension and lymph node metastasis remained significantly associated with the mutation. Our results are in line with previously reported data from the literature [8,30]. In their large meta-analysis, including 63 studies and 20,764 PTC patients with different ethnic and geographic backgrounds, Liu et al. [8] also reported a significant association between *BRAF*V600E mutation and extrathyroidal extension (*p* < 0.00001), *BRAF*V600E mutation and an advanced TNM stage (III/IV) (*p* < 0.00001), *BRAF*V600E mutation and lymph node metastasis (*p* < 0.00001), *BRAF*V600E mutation and tumor recurrence (*p* < 0.00001), respectively.

When looking at the impact of *BRAF*V600E mutation on patient’s outcomes and the occurrence of adverse events (recurrent/persistent disease, distant metastasis) we found that *BRAF*V600E mutation is valuable in the risk stratification assessment of PTC patients, yet not as an independent prognostic factor, but only when used in synergic interaction with other demographic and pathological risk factors.

In our study, both Kaplan–Meier and univariate analysis revealed a significant reduction of EFS among PTC patients with tumors harboring *BRAF*V600E mutation compared to PTC patients without mutation. Nevertheless, *BRAF*V600E mutation failed to remain an independent prognostic risk factor for PTC patients in multivariate analysis. By contrast, concurrent presence of *BRAF*V600E mutation with other risk factors (age ≥ 55 years old, male gender, conventional and tall cell histology, tumor size > 40 mm, extrathyroidal extension, multifocality and lymph node metastasis) resulted in being a better predictor of adverse outcomes for PTC patients in our study, compared to *BRAF*V600E mutation alone. Similar findings were already reported by previous studies [17,18,19,23]. Yet, our results add further evidence to support these studies.

The literature data are currently divided and highly controversial regarding the association between *BRAF*V600E mutation and poor prognosis in PTC. There are studies that have found *BRAF*V600E mutation to be an independent predictor of poor outcomes [8,12,31]. Xing et al., for example, in their large multicenter study including more than 2000 patients, demonstrated an independent prognostic value of *BRAF*V600E mutation for PTC recurrence even in patients with low TNM stage and micro-PTC [12]. Conversely, other authors have failed to demonstrate this [7,29,32,33]. In their recent systemic review including 11 studies and 4674 patients, Li. et al. [33] reported comparable rates of tumor recurrence between patients with PTC harboring *BRAF*V600E mutation and patients without mutation (HR 1.16, 95%CI 0.78–1.71). However, in a subgroup analysis, the authors found both geographical region and tumor stage as factors influencing the risk of recurrence associated with *BRAF*V600E mutation. These findings offer further support to the observation that heterogeneity of the data is relevant and should be considered when interpreting the impact of a *BRAF*V600E mutation on clinical outcomes [34].

Interestingly, when focusing on lower-risk patients with PTC (aged <55 years old, female, with tumors measuring ≤40 mm or with a single tumor foci), *BRAF*V600E mutation was strongly associated with a worse EFS in our study. Thus, in these subgroups of PTCs, *BRAF*V600E mutation could help to identify patients requiring more intensive treatment and follow-up. The potential role of *BRAF*V600E mutation as an aid to risk stratification in low-risk PTC patients (classified as such based on clinico-pathological criteria) has been an issue raised by others before. In a study focused on low-risk patients with intrathyroidal PTC (<4 cm, N0, M0) conducted by Elisei R et al. [35], *BRAF*V600E-mutated tumors had a recurrence rate of 8%, compared to only 1% in *BRAF*V600E wild-type tumors (*p* = 0.003, Fisher’s exact). These results offer some new, promising perspectives, but need to be further confirmed by additional studies.

In addition to clinico-pathological risk factors, co-existence of *BRAF*V600E mutation with other mutations (eg. *TERT* promoter mutations) has been shown to have promising prognostic value in PTC patients. In a large meta-analysis, Vuong HG et al. demonstrated that PTCs with co-existing *BRAF*V600E and *TERT* promoter mutations were associated with a significant increased risk for tumor aggressiveness and recurrence compared to PTCs with *BRAF* or *TERT* promoter mutations alone [36]. This highlights a key role for *BRAF*V600E mutation as a genetic driver for a higher mortality risk working in synergy with other genetic alterations, such as *TERT* promoter mutations. The role of *TERT* promoter and *BRAF*V600E mutation analysis has also been demonstrated for the diagnostic and prognostic evaluation of thyroid nodules processed with liquid-based cytology [37]. The expression of PD-L1 (programmed cell death ligand 1) on the other hand, has been identified as a valuable biomarker, associated with aggressiveness and negative outcomes in patients with thyroid cancer. Moreover, in some cases of advanced thyroid cancer, PD-L1 might be co-expressed with *BRAF*V600E and *TERT* promoter mutations, information that might have important consequences on patient prognosis and the possibility of using target therapies [38]. An understanding of the molecular mechanisms involved in the pathogenesis of differentiated thyroid cancer can also be useful to refine patient selection for radioiodine therapy. In their study, Pizzimeti C and el have nicely demonstrated that the assessment of *BRAF* mutations and AXL (Anexelekto thyrosine kinase receptor) expression could help identify patients who could be treated with higher-activity radioiodine therapy or with other possible therapies, such as immunotherapy, mainly those with higher expression of PD-L1 [39].

The oncogenic molecular mechanisms of *BRAF*V600E mutation in the pathogenesis of PTC and thyroid cancer in general are well documented in the literature. *BRAF*V600E mutation mimics phosphorylation in the active segment of *BRAF* leading to a constitutive activation of the kinase. As a result, *BRAF*V600E-driven tumors exhibit high extracellular signal-regulated kinase phosphorylation, leading to unregulated cell proliferation. The MAPK signaling inhibits at variable degree the expression of genes required for iodine uptake, which are hallmarks of the treatment of PTC [34,40]. Nevertheless, the mechanism associated with tumor aggressiveness in *BRAF*V600E-mutated PTCs remains unclear and probably other pathways cooperate to promote cancer progression [34]. Notch putative pathway, a highly conserved signaling pathway, crucial in development and with an important role in malignant transformation, might be implicated, as *BRAF*V600E mutation coupled with overexpression of the Notch intracellular domain leads to larger thyroid tumors, more aggressive disease and decreased overall survival [41]. Other pathways might be the overexpression of lysyl oxidase (LOX) [42] and the loss of individual SWI/SNF (switch/sucrose non-fermentable) subunits [43] that have been demonstrated as promoting disease progression and decrease survival in *BRAF*V600E-mutated tumors.

Our study has some limitations: the relatively small number of cases and the retrospective nature of the study, which might have caused a certain degree of selection bias. Yet, to the best of our knowledge, this is the first study addressing this topic in a Romanian population. Despite these limitations, our study covered a large period (between 2008–2015) and included all PTCs registered at our hospital that fulfilled the inclusion criteria. Moreover, we performed a complete morphological characterization and obtained relevant follow-up data for all PTC cases included in the study. In addition, the monocentric nature of the study is another limitation, as our data relies mainly on a single geographical region (Mures county). Nevertheless, as a university hospital, Târgu-Mureş Emergency Hospital provides medical services to patients coming from all over the country (including geographical regions located at greater distance), thus making the geographical bias less likely to have considerably affected our results.

## 5. Conclusions

To sum up, our results demonstrate the prognostic value of *BRAF*V600E mutation in the risk stratification assessment of PTC patients. Nevertheless, *BRAF*V600E mutation status should not be used alone, but rather should be integrated in the context of other clinicopathological risk factors. In our study, the synergic interaction between *BRAF*V600E mutation and age ≥55 years old, male gender, conventional and tall cell histology, tumor size >40 mm, extrathyroidal extension, multifocality and lymph node metastasis, respectively, resulted in being a better predictor of adverse outcomes for PTC patients compared to *BRAF*V600E mutation alone. *BRAF*V600E mutation status is also prognostically valuable among lower-risk subgroups of PTC patients (aged <55 years old, female, with tumors measuring ≤40 mm or with a single tumor foci), where it could help with identifying patients requiring more intensive treatment and follow-up. Further studies are needed to confirm this.

## Figures and Tables

**Figure 1 cancers-15-04053-f001:**
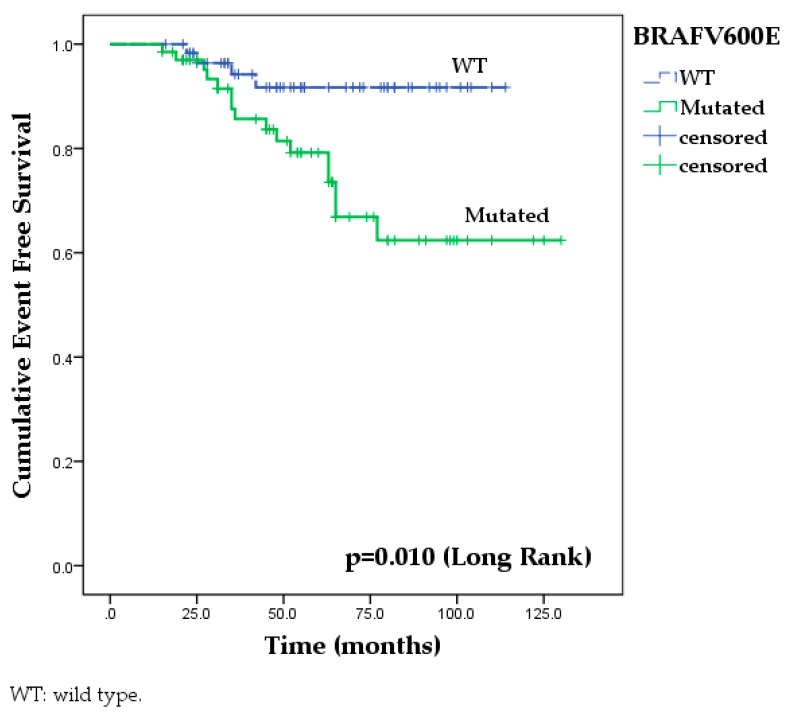
Kaplan–Meier analysis of the impact of *BRAF*V600E mutation on event-free survival of the total patients included in the study. The presence of BRAFV600E mutation was significantly associated with poor event-free survival among all study cases.

**Figure 2 cancers-15-04053-f002:**
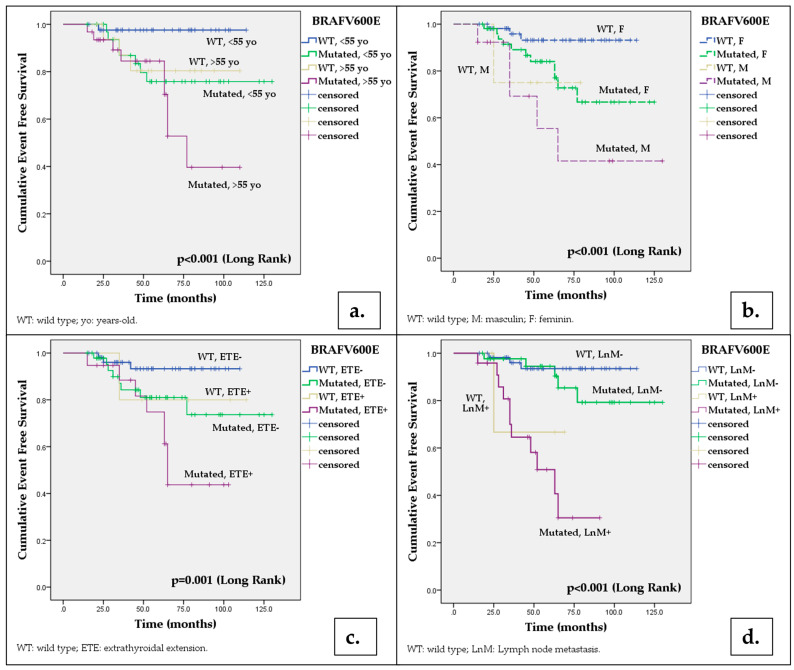
Kaplan–Meier analysis of the impact of *BRAF*V600E mutation associated with demographic and histological features on event-free survival of the total patients included in the study: (**a**) *BRAF*V600E mutation and patient’s age (<55 versus ≥55 years old); (**b**) *BRAF*V600E mutation and male gender; (**c**) *BRAF*V600E mutation and extrathyroidal extension; (**d**) *BRAF*V600E mutation and lymph node metastasis. *p*-values were obtained by applying the Log Rank test; multiple comparisons were performed with patients harboring wild-type *BRAF*V600E tumors, aged < 55 years old (**a**), of female gender (**b**), without extrathyroidal extension (**c**) and lymph node metastasis (**d**), respectively.

**Table 1 cancers-15-04053-t001:** Clinical and histopathological data for the study cases.

Factors	Total *n* = 127	*BRAF*V600E Wild-Type *n* = 60	*BRAF*V600E Mutated *n* = 67	*p* ^a^
Age at surgery (mean ± SD, years)	48.6 ± 1.28	46.18 ± 1.17	50.76 ± 1.72	0.075 *
Age (*n*, %)				** *0.037* **
<55 years	79 (62.2)	43 (71.7)	36 (53.7)	
≥55 years	48 (37.8)	17 (28.3)	31 (46.3)	
Gender, female (*n*, %)	110 (86.6)	56 (93.3)	54 (80.6)	** *0.035* **
Tumor size (mean ± SD, mm)	22.88 ± 1.5	23.35 ± 1.42	21.90 ± 1.33	0.458 *
Tumor size (*n*, %)				
11–20 mm	67 (52.8)	29 (48.3)	38 (56.7)	0.442
21–40 mm	51 (40.2)	27 (45.0)	24 (35.8)	0.225
>40 mm	9 (7.1)	4 (6.7)	5 (7.5)	0.864
Multifocality (*n*, %)	51 (40.2)	20 (33.3)	31 (46.3)	0.138
Histological variant (*n*, %)				
Conventional	88 (69.3)	32 (53.3)	56 (83.6)	** *0.0005* **
Tall cell variant	9 (7.1)	3 (5)	6 (9)	0.596
Warthin-like	7 (5.5)	3 (5)	4 (6)	0.886
Oncocytic	3 (2.4)	3 (5)	0	0.205
Solid	2 (1.6)	2 (3.3)	0	0.434
Follicular variant, infiltrative	13 (10.2)	12 (20)	1 (1.5)	** *0.001* **
Follicular variant, encapsulated, invasive	5 (3.9)	5 (8.3)	0	0.051
Extrathyroidal extension (*n*, %)	24 (18.9)	5 (8.3)	19 (28.4)	** *0.004* **
Primary tumor, pT (*n*, %)				
1b	51 (40.2)	25 (41.7)	26 (38.8)	0.879
2	44 (34.6)	25 (41.7)	19 (28.4)	0.165
3a	8 (6.3)	5 (8.3)	3 (4.5)	0.607
3b	24 (18.9)	5 (8.3)	19 (28.4)	** *0.007* **
Lymph node involvement (*n*, %)	26/39	2/8 (25)	24/31 (77.4)	** *0.001* **
Vascular invasion (*n*, %)	4 (3.3)	2 (3.1)	2 (3)	0.911
Positive surgical resection margin (*n*, %)	18 (14.2)	4 (6.7)	14 (20.9)	** *0.022* **
Stage grouping				
I	101 (79.5)	54 (90)	47 (70.1)	** *0.010* **
II	22 (17.3)	6 (10)	16 (23.9)	0.067
III	4 (3.1)	0	4 (6)	0.155
Type of surgery (*n*, %)				
Lobectomy	0	0	0	-
Total thyroidectomy	88 (69.3)	52 (86.7)	36 (53.7)	** *0.0001* **
Total thyroidectomy with lymph node dissection	39 (30.7)	8 (13.3)	31 (46.2)	** *0.0001* **
Follow-up data (*n*, median, months)	57 (CI: 9–130)	58 (CI: 17–114)	57 (CI: 9–130)	-
I^131^ therapy (*n*, %)	133 (100)	66 (100)	67 (100)	-
Disease free (*n*, %)	107 (84.3)	56 (93.4)	51 (76.1)	** *0.008* **
Persistent disease (*n*, %)	14 (11)	2 (3.3)	12 (17.9)	** *0.009* **
Recurrence (*n*, %)	6 (4.7)	2 (3.3)	4 (6)	0.484
Distant metastasis (*n*, %)	5 (3.9)	0	5 (7.5)	** *0.031* **

^a^ Either Chi-square or Fisher’s test (when the conditions of application of chi-square test were not met) were used; *p* value was obtained by comparing clinical and histopathological data from patients harboring wild-type *BRAF*V600E tumors to patients harboring *BRAF*V600E-mutated tumors. * Student’s *t* test was used in these two situations. Statistically significant differences are shown in bold and italics.

**Table 2 cancers-15-04053-t002:** Univariate and multivariate analysis of prognostic factors associated with *BRAF*V600E mutation for the study cases.

		Univariate Analysis			Multivariate Analysis		
Factors	*BRAF*V600E Positive (67)/N	OR	[95%CI]	*p*	OR	[95%CI]	*p*
Age ≥ 55 years	31/48	** *2.18* **	[1.04–4.56]	** *0.039* **	2.20	[0.90–5.37]	0.081
Sex, male	13/17	** *3.37* **	[1.03–10.98]	** *0.043* **	2.81	[0.67–11.72]	0.155
PTC, conventional	56/88	** *4.44* **	[1.95–10.13]	** *0.008* **	** *6.33* **	[2.18–18.40]	** *<0.001* **
PTC, “tall cell”	4/9	1.88	[0.44–7.82]	0.392			
Tumor size > 40 mm	5/9	1.13	[0.28–4.41]	0.861			
Extrathyroidal extension	19/24	** *4.35* **	[1.51–12.54]	** *0.006* **	** *5.83* **	[1.60–21.27]	** *0.007* **
Multifocality	31/51	1.72	[0.83–3.53]	0.139			
Lymph node metastasis	24/26	** *7.81* **	[2.52–24.20]	** *<0.001* **	** *4.77* **	[1.44–15.79]	** *0.010* **
Positive resection margin	14/18	** *3.69* **	[1.14–11.95]	** *0.028* **	2.25	[0.56–9.00]	0.249

OR—Odds ratio, 95%CI—95% confidence interval, *p* < 0.05, PTC—papillary thyroid carcinoma. Statistically significant differences are shown in bold and italics.

**Table 3 cancers-15-04053-t003:** Event-free survival at 12, 24, 48 and 60 months, respectively, for our study patients in relation to *BRAF*V600E mutation status and other demographic and pathological factors.

	*BRAF*V600E Wild Type				*BRAF*V600E Mutated				Log Rank	*p*
Factors	* 12 Months (%) (95%CI)	* 24 Months (%) (95%CI)	* 48 Months (%) (95%CI)	* 60 Months (%) (95%CI)	* 12 Months (%) (95%CI)	* 24 Months (%) (95%CI)	* 48 Months (%) (95%CI)	* 60 Months (%) (95%CI)		
Total	98.3 (96.6–100)	96.4 (93.9–98.4)	94.2 (90.9–97.5)	91.7 (87.7–95.7)	98.5 (97–100)	95.1 (92.3–97.9)	81.4 (76–86.8)	62.4 (54.2–70.6)	**6.581**	** *0.010* **
Age (n,%)										
<55 years	97.6 (95.2–100)	97.6 (95.2–100)	97.6 (95.2–100)	97.6 (95.2–100)	96.8 (96.4–100)	93.5 (89.1–97.8)	79.6 (72.1–87.1)	75.8 (67.8–83.8)	**5.314 ^a^**	** *0.021* **
≥55 years	93.8 (93.1–100)	93.8 (93.1–100)	80.4 (70.2–90.6)	80.4 (70.2–90.6)	96.8 (93.6–100)	93.4 (88.9–97.9)	84.5 (77.2–91.8)	52.8 (39.3–66.3)	**12.641 ^a^**	** *<0.001* **
Gender (*n*, %)										
Female	98.1 (96.3–100)	98.1 (96.3–100)	93.1 (89.2–97)	93.1 (89.2–97)	98.1 (96.2–100)	95.9 (93–98.8)	84 (78.4–89.6)	72.8 (65–80.6)	**5.274 ^b^**	** *0.022* **
Male	75 (53.3–96.7)	75 (53.3–96.7)	75 (53.3–96.7)	75 (53.3–96.7)	92.3 (84.9–100)	92.3 (84.9–100)	69.2 (47.7–78.1)	55.4 (38.1–72.7)	**13.427 ^b^**	** *<0.001* **
Histology										
PTC conventional	96.7 (93.4–100)	94.3 (90–98.6)	92.1 (87.2–97)	89.3 (83.7–94.9)	98.1 (96.2–100)	93.6 (90–97.2)	78.6 (72.2–85)	68.5 (59.8–77.2)	**7.973**	** *0.005* **
PTC tall cell variant	98.4 (96.8–100)	85.4 (81.5–89.3)	66.7 (39.5–93.9)	66.7 (39.5–93.9)	83.3 (68.1–98.5)	83.3 (68.1–96.5)	62.5 (41.2–83.8)	41.7 (19.5–63.9)	**5.997**	** *0.014* **
Tumor size										
≤40 mm	98.2(96.4–100)	96.2 (93.5–98.9)	91.2 (86.9–95.5)	91.2(86.9–95.5)	98.4(96.8–100)	96.7 (94.4–99)	81.7 (76.1–87.3)	68.3 (60.6–76)	**4.505 ^c^**	** *0.034* **
>40 mm	-	-	-	-	-	-	80 (62.1–97.9)	26.7 (4.1–49.3)	**8.221 ^c^**	** *0.004* **
Extrathyroidal extension										
Absent	98.1 (96.3–100)	96 (93.2–98.8)	93.3 (89.5–97.1)	93.3 (89.5–97.1)	97.8 (95.6–100)	95.2 (91.9–98.5)	81 (74.5–87.5)	73.7 (64.5–82.9)	3.608 ^d^	0.057
Present	99 (98–100)	80 (62.1–97.9)	80 (62.1–97.9)	80 (62.1–97.9)	94.7 (89.6–100)	88.4 (80.6–96.2)	74.8 (63.8–85.8)	43.7 (29.9–57.5)	**11.243 ^d^**	** *0.001* **
Multifocality										
Absent	97.2 (94.5–100)	97.2 (94.5–100)	97.2 (94.5–100)	97.2 (94.5–100)	97.2 (94.5–100)	97.2 (94.5–100)	85.8 (79.1–92.5)	74 (64.3–83.7)	**5.193 ^e^**	** *0.023* **
Prezent	95 (90–100)	95 (90–100)	88.7 (81.1–96.3)	81.8 (72.2–91.4)	96.7 (93.4–100)	92.9 (88.1–97.7)	77.4 (69.2–85.6)	59.7 (48.7–70.7)	**11.639 ^e^**	** *0.001* **
Lymph node metastases										
Absent	98.1 (96.3–100)	98.1 (96.3–100)	93.5 (89.8–97.2)	93.5 (89.8–97.2)	97.6 (95.2–100)	97.5 (95.2–100)	94.5 (90.7–98.3)	85.3 (92.4–78.2)	1.155 ^f^	0.282
Present	-	66.7(39.5–93.9)	66.7(39.5–93.9)	-	95.8 (91.7–100)	90.8 (84.6–97)	58.1 (46.6–69.6)	30.5 (17.2–43.8)	**23.84 ^f^**	** *<0.001* **

95%CI—95% confidence interval, *p* < 0.05; PTC—papillary thyroid carcinoma. * Percentage of patients without adverse events at that precise moment (event including persistent disease, recurrent disease, distant metastasis). ^a^ Log Rank obtained by comparison with patients aged < 55 years old and wild-type *BRAF*V600E tumors. ^b^ Log Rank obtained by comparison with female patients with wild-type *BRAF*V600E tumors. ^c^ Log Rank obtained by comparison with patients harboring wild-type *BRAF*V600E tumors ≤ 4 cm. ^d^ Log Rank obtained by comparison with patients harboring wild-type *BRAF*V600E tumors without extrathyroidal extension. ^e^ Log Rank obtained by comparison with patients harboring unifocal, wild-type *BRAF*V600E tumors. ^f^ Log Rank obtained by comparison with patients harboring wild-type *BRAF*V600E tumors and no lymph node involvement. The percentage of patients with EFS at 60 months was significantly reduced when *BRAF*V600E mutation was associated with age ≥ 55 years old (*p* < 0.001), male gender (*p* < 0.001), conventional (*p* = 0.005) and tall cell histology (*p* = 0.014), tumor size > 40 mm (*p* = 0.001), extrathyroidal extension (*p* = 0.001), multifocality (*p* = 0.001) and lymph node metastasis (*p* < 0.001). Statistically significant differences are shown in bold and italics.

**Table 4 cancers-15-04053-t004:** Univariate and multivariate analysis of prognostic factors for event-free survival among patients with PTC in our study.

		Univariate Analysis			Multivariate Analysis		
Factors	Event/N (Total = 127)	HR	[95%CI]	*p*	HR	[95%CI]	*p*
Age ≥ 55 years	12/48	** *2.74* **	[1.12–6.73]	** *0.027* **	2.39	[0.82–6.96]	0.109
Sex, male	6/17	** *3.91* **	[1.4–10.21]	** *0.005* **	** *3.80* **	[1.30–11.06]	** *0.014* **
PTC, conventional	12/88	0.82	[0.33–2.01]	0.620			
PTC, “tall cell” variant	4/9	2.71	[0.91–8.12]	0.075			
Tumor size > 40 mm	3/9	2.40	[0.70–8.23]	0.163			
Extrathyroidal extension	9/24	** *2.96* **	[1.22–7.15]	** *0.016* **	1.02	[0.31–3.47]	0.905
Multifocality	13/51	** *2.81* **	[1.12–7.05]	** *0.027* **	3.11	[0.97–9.95]	0.055
Lymph node metastasis	12/26	** *9.14* **	[3.63–22.97]	** *<0.001* **	** *6.71* **	[2.29–19.69]	** *<0.001* **
Positive resection margin	5/18	2.79	[0.99–7.89]	0.05			
Positive *BRAF*V600E mutation	16/67	** *3.74* **	[1.25–11.21]	** *0.018* **	1.02	[0.27–3.61]	0.998

HR—Hazard ratio, 95%CI—95% confidence interval, *p* < 0.05, PTC—papillary thyroid carcinoma. The event included persistent disease, recurrent disease, distant metastasis. Statistically significant differences are shown in bold and italics.

## Data Availability

The data presented in this study are available on request from the corresponding author. The data are not publicly available due to ethical restrictions (personal data protection of the patients included in the study).

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
