# Peer review of "Impact of BRAFV600E Mutation on Event-Free Survival in Patients with Papillary Thyroid Carcinoma: A Retrospective Study in a Romanian Population"

_cancers, 2023, doi:10.3390/cancers15164053_

Round 1

Reviewer 1 Report

Overall, this is an interesting analysis on the prognostic role of BRAFV600E mutation in a series of 127 papillary thyroid carcinomas and represents the first attempt to address this topic in a Romanian population. Literature data is controversial about the prognostic value of BRAFV600E mutation alone as an independent prognostic factor for papillary thyroid carcinoma. The authors showed that such a mutation should be integrated in the context of other clinicopathological risk factors. However, I would like the authors to clearly underline that they are not the first to make such an analysis and that their findings add further evidence to what has already been demonstrated by other authors. Moreover, I would like the authors to discuss further references about the role of BRAFV600E mutation in papillary thyroid carcinoma, for example considering manuscripts doi: 10.1002/cncy.22454, doi:10.3390/ijms241210024, doi:10.1111/cen.13413, doi:10.1007/s40618-023-02063-x. In addition, I suggest to discuss the monocentric nature of the study as a limitation of the analysis and to introduct the prognostic value of the integration of BRAFV600E mutation with mutations in other genes (for example TERT) as a future area of research. Lastly, I think that a punctuation and grammar check is needed. In addition, in line 305, there is a typo (CPT instead of PTC) and there is also an error in the page numbering (all result as page 2). 

English language needs a minor revision (for example you can use online programs to improve it).

Reviewer 2 Report

As the presence of the BRAFV600E mutation in PTC is still a doubtful topic in pathology, the most valuable aspect of your paper is that you have fluently and in detail described the usefulness of the BRAFV600E mutation in risk stratification of PTC patients. As a result, the work is publishable with a few small changes:

1. I suggest you to highlight in the Summary that, based on your findings, BRAFV600E could not be used as an independent predictive factor in PTC patients (Table 4). You may use some conclusions from the manuscript's end, which is written really nicely (conclussion section).

2. What are the p-values in Table 1 indicating? What was compared? As stated, the Fisher test was used. However, the minimum for using Fisher's test is table 2x2.

I believe it would be more acceptable to display one p-value for each factor (age, gender, tumor size,…), for example:

BRAFV600E wt

BRAFV600E mut

p

age <55

43

36

0.0362

age >= 55

17

31

3. The symbols for the Chi square test and the Student's T-test in Table 1 are identical (*). Please modify (use different symbols).

4. In Table 3, you mention only highly statistically significant results while omitting the other significant factors (with 0.02<p<0.05). Why? According to the table, p<0.05 is statistically significant. These sections should be rewritten or clarified.

5. Exact p-values should be removed from the discussion.
